# Semi-supervised Learning with Deep Generative Models

**Diederik P. Kingma**[*], **Danilo J. Rezende**[†], **Shakir Mohamed**[†], **Max Welling**[*]
[*]Machine Learning Group, Univ. of Amsterdam, {`D.P.Kingma, M.Welling`}@uva.nl
[†]Google Deepmind, {`danilor, shakir`}@google.com

## Abstract

The ever-increasing size of modern data sets combined with the difficulty of obtaining label information has made semi-supervised learning one of the problems of significant practical importance in modern data analysis. We revisit the approach to semi-supervised learning with generative models and develop new models that allow for effective generalisation from small labelled data sets to large unlabelled ones. Generative approaches have thus far been either inflexible, inefficient or non-scalable. We show that deep generative models and approximate Bayesian inference exploiting recent advances in variational methods can be used to provide significant improvements, making generative approaches highly competitive for semi-supervised learning.

## 1 Introduction

Semi-supervised learning considers the problem of classification when only a small subset of the observations have corresponding class labels. Such problems are of immense practical interest in a wide range of applications, including image search (Fergus et al., 2009), genomics (Shi and Zhang, 2011), natural language parsing (Liang, 2005), and speech analysis (Liu and Kirchhoff, 2013), where unlabelled data is abundant, but obtaining class labels is expensive or impossible to obtain for the entire data set. The question that is then asked is: how can properties of the data be used to improve decision boundaries and to allow for classification that is more accurate than that based on classifiers constructed using the labelled data alone. In this paper we answer this question by developing *probabilistic models for inductive and transductive semi-supervised learning* by utilising an explicit model of the data density, building upon recent advances in deep generative models and scalable variational inference (Kingma and Welling, 2014; Rezende et al., 2014).

Amongst existing approaches, the simplest algorithm for semi-supervised learning is based on a *self-training* scheme (Rosenberg et al., 2005) where the the model is bootstrapped with additional labelled data obtained from its own highly confident predictions; this process being repeated until some termination condition is reached. These methods are heuristic and prone to error since they can reinforce poor predictions. *Transductive SVMs* (TSVM) (Joachims, 1999) extend SVMs with the aim of max-margin classification while ensuring that there are as few unlabelled observations near the margin as possible. These approaches have difficulty extending to large amounts of unlabelled data, and efficient optimisation in this setting is still an open problem. *Graph-based methods* are amongst the most popular and aim to construct a graph connecting similar observations; label information propagates through the graph from labelled to unlabelled nodes by finding the minimum energy (MAP) configuration (Blum et al., 2004; Zhu et al., 2003). Graph-based approaches are sensitive to the graph structure and require eigen-analysis of the graph Laplacian, which limits the scale to which these methods can be applied – though efficient spectral methods are now available (Fergus et al., 2009). *Neural network*-based approaches combine unsupervised and supervised learning

---

For an updated version of this paper, please see `http://arxiv.org/abs/1406.5298`

by training feed-forward classifiers with an additional penalty from an auto-encoder or other unsupervised embedding of the data (Ranzato and Szummer, 2008; Weston et al., 2012). The Manifold Tangent Classifier (MTC) (Rifai et al., 2011) trains contrastive auto-encoders (CAEs) to learn the manifold on which the data lies, followed by an instance of TangentProp to train a classifier that is approximately invariant to local perturbations along the manifold. The idea of manifold learning using graph-based methods has most recently been combined with kernel (SVM) methods in the Atlas RBF model (Pitelis et al., 2014) and provides amongst most competitive performance currently available.

In this paper, we instead, choose to exploit the power of *generative models*, which recognise the semi-supervised learning problem as a specialised missing data imputation task for the classification problem. Existing generative approaches based on models such as Gaussian mixture or hidden Markov models (Zhu, 2006), have not been very successful due to the need for a large number of mixtures components or states to perform well. More recent solutions have used non-parametric density models, either based on trees (Kemp et al., 2003) or Gaussian processes (Adams and Ghahramani, 2009), but scalability and accurate inference for these approaches is still lacking. Variational approximations for semi-supervised clustering have also been explored previously (Li et al., 2009; Wang et al., 2009).

Thus, while a small set of generative approaches have been previously explored, a generalised and scalable probabilistic approach for semi-supervised learning is still lacking. It is this gap that we address through the following contributions:

- We describe a new framework for semi-supervised learning with generative models, employing rich parametric density estimators formed by the fusion of probabilistic modelling and deep neural networks.
- We show for the first time how variational inference can be brought to bear upon the problem of semi-supervised classification. In particular, we develop a stochastic variational inference algorithm that allows for joint optimisation of both model and variational parameters, and that is scalable to large datasets.
- We demonstrate the performance of our approach on a number of data sets providing state-of-the-art results on benchmark problems.
- We show qualitatively generative semi-supervised models learn to separate the data classes (content types) from the intra-class variabilities (styles), allowing in a very straightforward fashion to simulate analogies of images on a variety of datasets.

## 2 Deep Generative Models for Semi-supervised Learning

We are faced with data that appear as pairs $(\mathbf{X}, \mathbf{Y}) = \{(\mathbf{x}_1, y_1), \ldots, (\mathbf{x}_N, y_N)\}$, with the $i$-th observation $\mathbf{x}_i \in \mathbb{R}^D$ and the corresponding class label $y_i \in \{1, \ldots, L\}$. Observations will have corresponding latent variables, which we denote by $\mathbf{z}_i$. We will omit the index $i$ whenever it is clear that we are referring to terms associated with a single data point. In semi-supervised classification, only a subset of the observations have corresponding class labels; we refer to the empirical distribution over the labelled and unlabelled subsets as $\widetilde{p}_l(\mathbf{x}, y)$ and $\widetilde{p}_u(\mathbf{x})$, respectively. We now develop models for semi-supervised learning that exploit generative descriptions of the data to improve upon the classification performance that would be obtained using the labelled data alone.

**Latent-feature discriminative model (M1):** A commonly used approach is to construct a model that provides an embedding or feature representation of the data. Using these features, a separate classifier is thereafter trained. The embeddings allow for a clustering of related observations in a latent feature space that allows for accurate classification, even with a limited number of labels. Instead of a linear embedding, or features obtained from a regular auto-encoder, we construct a deep generative model of the data that is able to provide a more robust set of latent features. The generative model we use is:

$$p(\mathbf{z}) = \mathcal{N}(\mathbf{z}|\mathbf{0}, \mathbf{I}); \qquad p_\theta(\mathbf{x}|\mathbf{z}) = f(\mathbf{x}; \mathbf{z}, \boldsymbol{\theta}), \tag{1}$$

where $f(\mathbf{x}; \mathbf{z}, \boldsymbol{\theta})$ is a suitable likelihood function (e.g., a Gaussian or Bernoulli distribution) whose probabilities are formed by a non-linear transformation, with parameters $\boldsymbol{\theta}$, of a set of latent variables $\mathbf{z}$. This non-linear transformation is essential to allow for higher moments of the data to be captured by the density model, and we choose these non-linear functions to be deep neural networks.

Approximate samples from the posterior distribution over the latent variables $p(\mathbf{z}|\mathbf{x})$ are used as features to train a classifier that predicts class labels $y$, such as a (transductive) SVM or multinomial regression. Using this approach, we can now perform classification in a lower dimensional space since we typically use latent variables whose dimensionality is much less than that of the observations. These low dimensional embeddings should now also be more easily separable since we make use of independent latent Gaussian posteriors whose parameters are formed by a sequence of non-linear transformations of the data. This simple approach results in improved performance for SVMs, and we demonstrate this in section 4.

**Generative semi-supervised model (M2):** We propose a probabilistic model that describes the data as being generated by a latent class variable $y$ in addition to a continuous latent variable $\mathbf{z}$. The data is explained by the generative process:

$$p(y) = \text{Cat}(y|\boldsymbol{\pi}); \qquad p(\mathbf{z}) = \mathcal{N}(\mathbf{z}|\mathbf{0}, \mathbf{I}); \qquad p_\theta(\mathbf{x}|y, \mathbf{z}) = f(\mathbf{x}; y, \mathbf{z}, \boldsymbol{\theta}), \qquad (2)$$

where $\text{Cat}(y|\boldsymbol{\pi})$ is the multinomial distribution, the class labels $y$ are treated as latent variables if no class label is available and $\mathbf{z}$ are additional latent variables. These latent variables are marginally independent and allow us, in case of digit generation for example, to separate the class specification from the writing style of the digit. As before, $f(\mathbf{x}; y, \mathbf{z}, \boldsymbol{\theta})$ is a suitable likelihood function, e.g., a Bernoulli or Gaussian distribution, parameterised by a non-linear transformation of the latent variables. In our experiments, we choose deep neural networks as this non-linear function. Since most labels $y$ are unobserved, we integrate over the class of any unlabelled data during the inference process, thus performing classification as inference. Predictions for any missing labels are obtained from the inferred posterior distribution $p_\theta(y|\mathbf{x})$. This model can also be seen as a hybrid continuous-discrete mixture model where the different mixture components share parameters.

**Stacked generative semi-supervised model (M1+M2):** We can combine these two approaches by first learning a new latent representation $\mathbf{z}_1$ using the generative model from M1, and subsequently learning a generative semi-supervised model M2, using embeddings from $\mathbf{z}_1$ instead of the raw data $\mathbf{x}$. The result is a deep generative model with two layers of stochastic variables: $p_\theta(\mathbf{x}, y, \mathbf{z}_1, \mathbf{z}_2) = p(y)p(\mathbf{z}_2)p_\theta(\mathbf{z}_1|y, \mathbf{z}_2)p_\theta(\mathbf{x}|\mathbf{z}_1)$, where the priors $p(y)$ and $p(\mathbf{z}_2)$ equal those of $y$ and $\mathbf{z}$ above, and both $p_\theta(\mathbf{z}_1|y, \mathbf{z}_2)$ and $p_\theta(\mathbf{x}|\mathbf{z}_1)$ are parameterised as deep neural networks.

## 3 Scalable Variational Inference

### 3.1 Lower Bound Objective

In all our models, computation of the exact posterior distribution is intractable due to the nonlinear, non-conjugate dependencies between the random variables. To allow for tractable and scalable inference and parameter learning, we exploit recent advances in variational inference (Kingma and Welling, 2014; Rezende et al., 2014). For all the models described, we introduce a fixed-form distribution $q_\phi(\mathbf{z}|\mathbf{x})$ with parameters $\phi$ that approximates the true posterior distribution $p(\mathbf{z}|\mathbf{x})$. We then follow the variational principle to derive a lower bound on the marginal likelihood of the model – this bound forms our objective function and ensures that our approximate posterior is as close as possible to the true posterior.

We construct the approximate posterior distribution $q_\phi(\cdot)$ as an inference or recognition model, which has become a popular approach for efficient variational inference (Dayan, 2000; Kingma and Welling, 2014; Rezende et al., 2014; Stuhlmüller et al., 2013). Using an inference network, we avoid the need to compute per data point variational parameters, but can instead compute a set of global variational parameters $\phi$. This allows us to amortise the cost of inference by generalising between the posterior estimates for all latent variables through the parameters of the inference network, and allows for fast inference at both training and testing time (unlike with VEM, in which we repeat the generalized E-step optimisation for every test data point). An inference network is introduced for all latent variables, and we parameterise them as deep neural networks whose outputs form the parameters of the distribution $q_\phi(\cdot)$. For the latent-feature discriminative model (M1), we use a Gaussian inference network $q_\phi(\mathbf{z}|\mathbf{x})$ for the latent variable $\mathbf{z}$. For the generative semi-supervised model (M2), we introduce an inference model for each of the latent variables $\mathbf{z}$ and $y$, which we we assume has a factorised form $q_\phi(\mathbf{z}, y|\mathbf{x}) = q_\phi(\mathbf{z}|\mathbf{x})q_\phi(y|\mathbf{x})$, specified as Gaussian and multinomial distributions respectively.

$$\text{M1: } q_\phi(\mathbf{z}|\mathbf{x}) = \mathcal{N}(\mathbf{z}|\boldsymbol{\mu}_\phi(\mathbf{x}), \text{diag}(\boldsymbol{\sigma}_\phi^2(\mathbf{x}))), \qquad (3)$$

$$\text{M2: } q_\phi(\mathbf{z}|y, \mathbf{x}) = \mathcal{N}(\mathbf{z}|\boldsymbol{\mu}_\phi(y, \mathbf{x}), \text{diag}(\boldsymbol{\sigma}_\phi^2(\mathbf{x}))); \quad q_\phi(y|\mathbf{x}) = \text{Cat}(y|\boldsymbol{\pi}_\phi(\mathbf{x})), \qquad (4)$$

where $\boldsymbol{\sigma}_\phi(\mathbf{x})$ is a vector of standard deviations, $\boldsymbol{\pi}_\phi(\mathbf{x})$ is a probability vector, and the functions $\boldsymbol{\mu}_\phi(\mathbf{x}), \boldsymbol{\sigma}_\phi(\mathbf{x})$ and $\boldsymbol{\pi}_\phi(\mathbf{x})$ are represented as MLPs.

### 3.1.1 Latent Feature Discriminative Model Objective

For this model, the variational bound $\mathcal{J}(\mathbf{x})$ on the marginal likelihood for a single data point is:

$$\log p_\theta(\mathbf{x}) \geq \mathbb{E}_{q_\phi(\mathbf{z}|\mathbf{x})}\left[\log p_\theta(\mathbf{x}|\mathbf{z})\right] - KL[q_\phi(\mathbf{z}|\mathbf{x})\|p_\theta(\mathbf{z})] = -\mathcal{J}(\mathbf{x}), \tag{5}$$

The inference network $q_\phi(\mathbf{z}|\mathbf{x})$ (3) is used during training of the model using both the labelled and unlabelled data sets. This approximate posterior is then used as a feature extractor for the labelled data set, and the features used for training the classifier.

### 3.1.2 Generative Semi-supervised Model Objective

For this model, we have two cases to consider. In the first case, the label corresponding to a data point is observed and the variational bound is a simple extension of equation (5):

$$\log p_\theta(\mathbf{x}, y) \geq \mathbb{E}_{q_\phi(\mathbf{z}|\mathbf{x}, y)}\left[\log p_\theta(\mathbf{x}|y, \mathbf{z}) + \log p_\theta(y) + \log p(\mathbf{z}) - \log q_\phi(\mathbf{z}|\mathbf{x}, y)\right] = -\mathcal{L}(\mathbf{x}, y), \tag{6}$$

For the case where the label is missing, it is treated as a latent variable over which we perform posterior inference and the resulting bound for handling data points with an unobserved label $y$ is:

$$\begin{aligned}\log p_\theta(\mathbf{x}) &\geq \mathbb{E}_{q_\phi(y,\mathbf{z}|\mathbf{x})}\left[\log p_\theta(\mathbf{x}|y, \mathbf{z}) + \log p_\theta(y) + \log p(\mathbf{z}) - \log q_\phi(y, \mathbf{z}|\mathbf{x})\right] \\ &= \sum_y q_\phi(y|\mathbf{x})(-\mathcal{L}(\mathbf{x}, y)) + \mathcal{H}(q_\phi(y|\mathbf{x})) = -\mathcal{U}(\mathbf{x}).\end{aligned} \tag{7}$$

The bound on the marginal likelihood for the entire dataset is now:

$$\mathcal{J} = \sum_{(\mathbf{x}, y) \sim \widetilde{p}_l} \mathcal{L}(\mathbf{x}, y) + \sum_{\mathbf{x} \sim \widetilde{p}_u} \mathcal{U}(\mathbf{x}) \tag{8}$$

The distribution $q_\phi(y|\mathbf{x})$ (4) for the missing labels has the form a discriminative classifier, and we can use this knowledge to construct the best classifier possible as our inference model. This distribution is also used at test time for predictions of any unseen data.

In the objective function (8), the label predictive distribution $q_\phi(y|\mathbf{x})$ contributes only to the second term relating to the unlabelled data, which is an undesirable property if we wish to use this distribution as a classifier. Ideally, all model and variational parameters should learn in all cases. To remedy this, we add a classification loss to (8), such that the distribution $q_\phi(y|\mathbf{x})$ also learns from labelled data. The extended objective function is:

$$\mathcal{J}^\alpha = \mathcal{J} + \alpha \cdot \mathbb{E}_{\widetilde{p}_l(\mathbf{x}, y)}\left[-\log q_\phi(y|\mathbf{x})\right], \tag{9}$$

where the hyper-parameter $\alpha$ controls the relative weight between generative and purely discriminative learning. We use $\alpha = 0.1 \cdot N$ in all experiments. While we have obtained this objective function by motivating the need for all model components to learn at all times, the objective 9 can also be derived directly using the variational principle by instead performing inference over the parameters $\boldsymbol{\pi}$ of the categorical distribution, using a symmetric Dirichlet prior over these parameterss.

## 3.2 Optimisation

The bounds in equations (5) and (9) provide a unified objective function for optimisation of both the parameters $\boldsymbol{\theta}$ and $\boldsymbol{\phi}$ of the generative and inference models, respectively. This optimisation can be done jointly, without resort to the variational EM algorithm, by using deterministic reparameterisations of the expectations in the objective function, combined with Monte Carlo approximation – referred to in previous work as *stochastic gradient variational Bayes* (SGVB) (Kingma and Welling, 2014) or as *stochastic backpropagation* (Rezende et al., 2014). We describe the core strategy for the latent-feature discriminative model M1, since the same computations are used for the generative semi-supervised model.

When the prior $p(\mathbf{z})$ is a spherical Gaussian distribution $p(\mathbf{z}) = \mathcal{N}(\mathbf{z}|\mathbf{0}, \mathbf{I})$ and the variational distribution $q_\phi(\mathbf{z}|\mathbf{x})$ is also a Gaussian distribution as in (3), the KL term in equation (5) can be computed

---
**Algorithm 1** Learning in model M1
---

> **while** generativeTraining() **do**
>   $\mathcal{D} \leftarrow$ getRandomMiniBatch()
>   $\mathbf{z}_i \sim q_\phi(\mathbf{z}_i|\mathbf{x}_i) \quad \forall \mathbf{x}_i \in \mathcal{D}$
>   $\mathcal{J} \leftarrow \sum_n \mathcal{J}(\mathbf{x}_i)$
>   $(\mathbf{g}_\theta, \mathbf{g}_\phi) \leftarrow (\frac{\partial \mathcal{J}}{\partial \theta}, \frac{\partial \mathcal{J}}{\partial \phi})$
>   $(\boldsymbol{\theta}, \boldsymbol{\phi}) \leftarrow (\boldsymbol{\theta}, \boldsymbol{\phi}) + \boldsymbol{\Gamma}(\mathbf{g}_\theta, \mathbf{g}_\phi)$
> **end while**
> **while** discriminativeTraining() **do**
>   $\mathcal{D} \leftarrow$ getLabeledRandomMiniBatch()
>   $\mathbf{z}_i \sim q_\phi(\mathbf{z}_i|\mathbf{x}_i) \quad \forall \{\mathbf{x}_i, y_i\} \in \mathcal{D}$
>   trainClassifier($\{\mathbf{z}_i, y_i\}$ )
> **end while**

---
**Algorithm 2** Learning in model M2
---

> **while** training() **do**
>   $\mathcal{D} \leftarrow$ getRandomMiniBatch()
>   $y_i \sim q_\phi(y_i|\mathbf{x}_i) \quad \forall \{\mathbf{x}_i, y_i\} \notin \mathcal{O}$
>   $\mathbf{z}_i \sim q_\phi(\mathbf{z}_i|y_i, \mathbf{x}_i)$
>   $\mathcal{J}^\alpha \leftarrow$ eq. (9)
>   $(\mathbf{g}_\theta, \mathbf{g}_\phi) \leftarrow (\frac{\partial \mathcal{L}^\alpha}{\partial \theta}, \frac{\partial \mathcal{L}^\alpha}{\partial \phi})$
>   $(\boldsymbol{\theta}, \boldsymbol{\phi}) \leftarrow (\boldsymbol{\theta}, \boldsymbol{\phi}) + \boldsymbol{\Gamma}(\mathbf{g}_\theta, \mathbf{g}_\phi)$
> **end while**

analytically and the log-likelihood term can be rewritten, using the location-scale transformation for the Gaussian distribution, as:

$$\mathbb{E}_{q_\phi(\mathbf{z}|\mathbf{x})}\left[\log p_\theta(\mathbf{x}|\mathbf{z})\right] = \mathbb{E}_{\mathcal{N}(\boldsymbol{\epsilon}|\mathbf{0},\mathbf{I})}\left[\log p_\theta(\mathbf{x}|\boldsymbol{\mu}_\phi(\mathbf{x}) + \boldsymbol{\sigma}_\phi(\mathbf{x}) \odot \boldsymbol{\epsilon})\right], \tag{10}$$

where $\odot$ indicates the element-wise product. While the expectation (10) still cannot be solved analytically, its gradients with respect to the generative parameters $\boldsymbol{\theta}$ and variational parameters $\boldsymbol{\phi}$ can be efficiently computed as expectations of simple gradients:

$$\nabla_{\{\theta,\phi\}}\mathbb{E}_{q_\phi(\mathbf{z}|\mathbf{x})}\left[\log p_\theta(\mathbf{x}|\mathbf{z})\right] = \mathbb{E}_{\mathcal{N}(\boldsymbol{\epsilon}|\mathbf{0},\mathbf{I})}\left[\nabla_{\{\theta,\phi\}}\log p_\theta(\mathbf{x}|\boldsymbol{\mu}_\phi(\mathbf{x}) + \boldsymbol{\sigma}_\phi(\mathbf{x}) \odot \boldsymbol{\epsilon})\right]. \tag{11}$$

The gradients of the loss (9) for model M2 can be computed by a direct application of the chain rule and by noting that the conditional bound $\mathcal{L}(\mathbf{x}_n, y)$ contains the same type of terms as the loss (9). The gradients of the latter can then be efficiently estimated using (11) .

During optimization we use the estimated gradients in conjunction with standard stochastic gradient-based optimization methods such as SGD, RMSprop or AdaGrad (Duchi et al., 2010). This results in parameter updates of the form: $(\boldsymbol{\theta}^{t+1}, \boldsymbol{\phi}^{t+1}) \leftarrow (\boldsymbol{\theta}^t, \boldsymbol{\phi}^t) + \boldsymbol{\Gamma}^t(\mathbf{g}_\theta^t, \mathbf{g}_\phi^t)$, where $\boldsymbol{\Gamma}$ is a diagonal preconditioning matrix that adaptively scales the gradients for faster minimization. The training procedure for models M1 and M2 are summarised in algorithms 1 and 2, respectively. Our experimental results were obtained using AdaGrad.

### 3.3 Computational Complexity

The overall algorithmic complexity of a single joint update of the parameters $(\boldsymbol{\theta}, \boldsymbol{\phi})$ for M1 using the estimator (11) is $C_{\mathbf{M1}} = MSC_{\text{MLP}}$ where $M$ is the minibatch size used , $S$ is the number of samples of the random variate $\boldsymbol{\epsilon}$, and $C_{\text{MLP}}$ is the cost of an evaluation of the MLPs in the conditional distributions $p_\theta(\mathbf{x}|\mathbf{z})$ and $q_\phi(\mathbf{z}|\mathbf{x})$. The cost $C_{\text{MLP}}$ is of the form $O(KD^2)$ where $K$ is the total number of layers and $D$ is the average dimension of the layers of the MLPs in the model. Training M1 also requires training a supervised classifier, whose algorithmic complexity, if it is a neural net, it will have a complexity of the form $C_{\text{MLP}}$ .

The algorithmic complexity for M2 is of the form $C_{\text{M2}} = LC_{\text{M1}}$, where $L$ is the number of labels and $C_{\text{M1}}$ is the cost of evaluating the gradients of each conditional bound $\mathcal{J}_y(x)$, which is the same as for M1. The stacked generative semi-supervised model has an algorithmic complexity of the form $C_{\mathbf{M1}} + C_{\mathbf{M2}}$. But with the advantage that the cost $C_{\mathbf{M2}}$ is calculated in a low-dimensional space (formed by the latent variables of the model M1 that provides the embeddings).

These complexities make this approach extremely appealing, since they are no more expensive than alternative approaches based on auto-encoder or neural models, which have the lowest computational complexity amongst existing competitive approaches. In addition, our models are fully probabilistic, allowing for a wide range of inferential queries, which is not possible with many alternative approaches for semi-supervised learning.

Table 1: Benchmark results of semi-supervised classification on MNIST with few labels.

| $N$ | NN | CNN | TSVM | CAE | MTC | AtlasRBF | M1+TSVM | M2 | M1+M2 |
|------|-------|-------|-------|------|------|-----------------|------------------|------------------|------------------|
| 100  | 25.81 | 22.98 | 16.81 | 13.47 | 12.03 | 8.10 ($\pm$ 0.95) | 11.82 ($\pm$ 0.25) | 11.97 ($\pm$ 1.71) | **3.33** ($\pm$ 0.14) |
| 600  | 11.44 | 7.68  | 6.16  | 6.3  | 5.13 | –               | 5.72 ($\pm$ 0.049) | 4.94 ($\pm$ 0.13) | **2.59** ($\pm$ 0.05) |
| 1000 | 10.7  | 6.45  | 5.38  | 4.77 | 3.64 | 3.68 ($\pm$ 0.12) | 4.24 ($\pm$ 0.07) | 3.60 ($\pm$ 0.56) | **2.40** ($\pm$ 0.02) |
| 3000 | 6.04  | 3.35  | 3.45  | 3.22 | 2.57 | –               | 3.49 ($\pm$ 0.04) | 3.92 ($\pm$ 0.63) | **2.18** ($\pm$ 0.04) |

## 4 Experimental Results

Open source code, with which the most important results and figures can be reproduced, is available at `http://github.com/dpkingma/nips14-ssl`. For the latest experimental results, please see `http://arxiv.org/abs/1406.5298`.

### 4.1 Benchmark Classification

We test performance on the standard MNIST digit classification benchmark. The data set for semi-supervised learning is created by splitting the 50,000 training points between a labelled and unlabelled set, and varying the size of the labelled from 100 to 3000. We ensure that all classes are balanced when doing this, i.e. each class has the same number of labelled points. We create a number of data sets using randomised sampling to confidence bounds for the mean performance under repeated draws of data sets.

For model M1 we used a 50-dimensional latent variable $\mathbf{z}$. The MLPs that form part of the generative and inference models were constructed with two hidden layers, each with 600 hidden units, using softplus $\log(1+e^x)$ activation functions. On top, a transductive SVM (TSVM) was learned on values of $\mathbf{z}$ inferred with $q_\phi(\mathbf{z}|\mathbf{x})$. For model M2 we also used 50-dimensional $\mathbf{z}$. In each experiment, the MLPs were constructed with one hidden layer, each with 500 hidden units and softplus activation functions. In case of SVHN and NORB, we found it helpful to pre-process the data with PCA. This makes the model one level deeper, and still optimizes a lower bound on the likelihood of the unprocessed data.

Table 1 shows classification results. We compare to a broad range of existing solutions in semi-supervised learning, in particular to classification using nearest neighbours (NN), support vector machines on the labelled set (SVM), the transductive SVM (TSVM), and contractive auto-encoders (CAE). Some of the best results currently are obtained by the manifold tangent classifier (MTC) (Rifai et al., 2011) and the AtlasRBF method (Pitelis et al., 2014). Unlike the other models in this comparison, our models are fully probabilistic but have a cost in the same order as these alternatives.

**Results:** The latent-feature discriminative model (M1) performs better than other models based on simple embeddings of the data, demonstrating the effectiveness of the latent space in providing robust features that allow for easier classification. By combining these features with a classification mechanism directly in the same model, as in the conditional generative model (M2), we are able to get similar results without a separate TSVM classifier.

However, by far the best results were obtained using the stack of models M1 and M2. This combined model provides accurate test-set predictions across all conditions, and easily outperforms the previously best methods. We also tested this deep generative model for supervised learning with all available labels, and obtain a test-set performance of 0.96%, which is among the best published results for this permutation-invariant MNIST classification task.

### 4.2 Conditional Generation

The conditional generative model can be used to explore the underlying structure of the data, which we demonstrate through two forms of analogical reasoning. Firstly, we demonstrate style and content separation by fixing the class label $y$, and then varying the latent variables $\mathbf{z}$ over a range of values. Figure 1 shows three MNIST classes in which, using a trained model with two latent variables, and the 2D latent variable varied over a range from -5 to 5. In all cases, we see that nearby regions of latent space correspond to similar writing styles, independent of the class; the left region represents upright writing styles, while the right-side represents slanted styles.

As a second approach, we use a test image and pass it through the inference network to infer a value of the latent variables corresponding to that image. We then fix the latent variables $\mathbf{z}$ to this

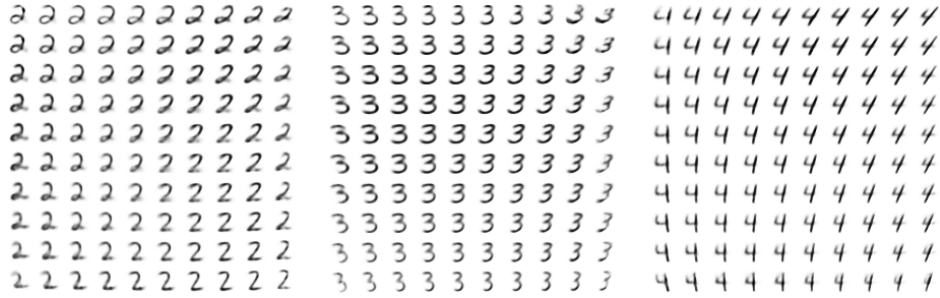

(a) Handwriting styles for MNIST obtained by fixing the class label and varying the 2D latent variable **z**

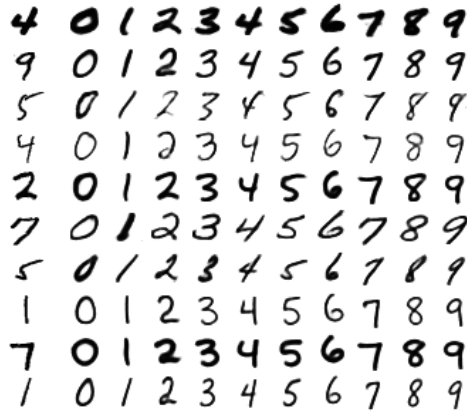

(b) MNIST analogies

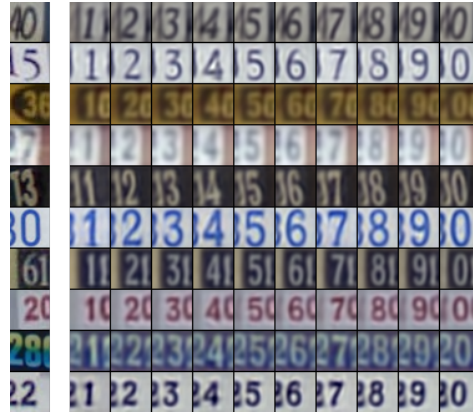

(c) SVHN analogies

Figure 1: **(a)** Visualisation of handwriting styles learned by the model with 2D **z**-space. **(b,c)** Analogical reasoning with generative semi-supervised models using a high-dimensional **z**-space. The leftmost columns show images from the test set. The other columns show analogical fantasies of **x** by the generative model, where the latent variable **z** of each row is set to the value inferred from the test-set image on the left by the inference network. Each column corresponds to a class label **y**.

Table 2: Semi-supervised classification on the SVHN dataset with 1000 labels.

| KNN | TSVM | M1+KNN | M1+TSVM | M1+M2 |
|---|---|---|---|---|
| 77.93 | 66.55 | 65.63 | 54.33 | **36.02** |
| (± 0.08) | (± 0.10) | (± 0.15) | (± 0.11) | (± 0.10) |

Table 3: Semi-supervised classification on the NORB dataset with 1000 labels.

| KNN | TSVM | M1+KNN | M1+TSVM |
|---|---|---|---|
| 78.71 | 26.00 | 65.39 | **18.79** |
| (± 0.02) | (± 0.06) | (± 0.09) | (± 0.05) |

value, vary the class label **y**, and simulate images from the generative model corresponding to that combination of **z** and **y**. This again demonstrate the disentanglement of style from class. Figure 1 shows these analogical fantasies for the MNIST and SVHN datasets (Netzer et al., 2011). The SVHN data set is a far more complex data set than MNIST, but the model is able to fix the style of house number and vary the digit that appears in that style well. These generations represent the best current performance in simulation from generative models on these data sets.

The model used in this way also provides an alternative model to the stochastic feed-forward networks (SFNN) described by Tang and Salakhutdinov (2013). The performance of our model significantly improves on SFNN, since instead of an inefficient Monte Carlo EM algorithm relying on importance sampling, we are able to perform efficient joint inference that is easy to scale.

### 4.3 Image Classification

We demonstrate the performance of image classification on the SVHN, and NORB image data sets. Since no comparative results in the semi-supervised setting exists, we perform nearest-neighbour and TSVM classification with RBF kernels and compare performance on features generated by our latent-feature discriminative model to the original features. The results are presented in tables 2 and 3, and we again demonstrate the effectiveness of our approach for semi-supervised classification.

### 4.4 Optimization details

The parameters were initialized by sampling randomly from $\mathcal{N}(\mathbf{0}, 0.001^2\mathbf{I})$, except for the bias parameters which were initialized as 0. The objectives were optimized using minibatch gradient ascent until convergence, using a variant of RMSProp with momentum and initialization bias correction, a constant learning rate of $0.0003$, first moment decay (momentum) of $0.1$, and second moment decay of $0.001$. For MNIST experiments, minibatches for training were generated by treating normalised pixel intensities of the images as Bernoulli probabilities and sampling binary images from this distribution. In the M2 model, a weight decay was used corresponding to a prior of $(\boldsymbol{\theta}, \boldsymbol{\phi}) \sim \mathcal{N}(0, I)$.

## 5 Discussion and Conclusion

The approximate inference methods introduced here can be easily extended to the model's parameters, harnessing the full power of variational learning. Such an extension also provides a principled ground for performing model selection. Efficient model selection is particularly important when the amount of available data is not large, such as in semi-supervised learning.

For image classification tasks, one area of interest is to combine such methods with convolutional neural networks that form the gold-standard for current supervised classification methods. Since all the components of our model are parametrised by neural networks we can readily exploit convolutional or more general locally-connected architectures – and forms a promising avenue for future exploration.

A limitation of the models we have presented is that they scale linearly in the number of classes in the data sets. Having to re-evaluate the generative likelihood for each class during training is an expensive operation. Potential reduction of the number of evaluations could be achieved by using a truncation of the posterior mass. For instance we could combine our method with the truncation algorithm suggested by Pal et al. (2005), or by using mechanisms such as error-correcting output codes (Dietterich and Bakiri, 1995). The extension of our model to multi-label classification problems that is essential for image-tagging is also possible, but requires similar approximations to reduce the number of likelihood-evaluations per class.

We have developed new models for semi-supervised learning that allow us to improve the quality of prediction by exploiting information in the data density using generative models. We have developed an efficient variational optimisation algorithm for approximate Bayesian inference in these models and demonstrated that they are amongst the most competitive models currently available for semi-supervised learning. We hope that these results stimulate the development of even more powerful semi-supervised classification methods based on generative models, of which there remains much scope.

**Acknowledgements.** We are grateful for feedback from the reviewers. We would also like to thank the SURFFoundation for the use of the Dutch national e-infrastructure for a significant part of the experiments.

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
