[Reviews · NeurIPS 2014]

Submitted by Assigned_Reviewer_2

This paper builds on the recently introduced variational autoencoder (VAE) framework of Kingma and Welling (2014) and of Rezende et al. (2014) to introduce two separate semi-supervised learning models. The first (called M1) is a VAE version of self-taught learning (Raina et al., 2007), while the second (M2) blends a standard discriminative model with a conditional
generative VAE that models the input data given the class label.

I am conflicted about this paper. On the positive side I find the VAE a very interesting and important modeling framework and I find the idea of extending this framework to the semi-supervised domain to be both natural and sensible. I would expect this to be the first in a long series of papers exploring this direction. On the negative side, the specific two proposed approaches seem fairly straightforward. This isn't a strong criticism but it does impact the novelty of the paper. It would have been better if the experiments really focused on the differences between the two, in order to give the reader some idea of their differential abilities, but that wasn't really the case as they were only really compared directly in one set of experiments limited to the MNIST hand-written digits database. More seriously, there are a large number of similar methods that
are not discussed and not compared empirically. In particular there are many papers not cited discussing learning in the self-taught learning framework (including the original self-taught learning paper of Raina et al. (2007)) which is essentially identical to the proposed M1 approach. Other missing citations and comparisons are discussed below in
more specific comments on the experiments.

The empirical evaluation of the proposed approach is based on three types of experiments:

1 - Benchmark Classification: The first is a basic comparison of the proposed approach in a semi-supervised version of the MNIST hand-written digits database. These results are based entirely on the experiments of Rifai et al. (2011) where the authors have simply added the results for the proposed approach. This isn't really a problem, except that they neglect to provide all of the results from Rifai et al. (2011). While they include the results for 100, 600 and 1000 labeled examples. They do not provide results for either 3000 or 5000 (all) labeled examples. Why do they neglect to provide these 2 numbers? I think it's important to see how the proposed methods (M1 and M2) work in the larger labeled set regime.

2 - Conditional Generation: The second set of experiments explore the ability of the model to separately model label content and other more stylistic variations in two datasets: SVHN (street-view house numbers) and the NORB object dataset. Here the authors provide a set of qualitative results -- sampled images with variable "style" and "content" (label info). These results are
mildly compelling in that they capture -- for SVHN -- the color information of both the background and foreground but do not seem to capture the font style. There are also several papers discussing the separation of style and content that are either cited nor compared. These include: Tenenbaum and Freeman (2000), "Separating Style and Content with Bilinear Models"; Grimes and Rao (2005) "Bilinear sparse coding for invariant vision"; Rifai et al (2012) "Disentangling factors of variation for facial expression recognition"; Desjardins et al. (2012) "Disentangling Factors of Variation via Generative Entangling" and Reed et al. (2014) "Learning to Disentangle Factors of Variation with Manifold Interaction"

3 - Image Classification: The third set of experiments explore semi-supervised classification with SVHN and NORB image datasets. The authors state that there are no published results on these datasets in the semi-supervised setting so they compare to their own results using nearest neighbors and Transductive SVM. While I know of no results on these two
datasets in the semi-supervised regime, there are plenty of results on other image classification datasets with published results in the semi-supervised regime. Examples include: CIFAR-10 object recognition
dataset, Caltech-101, and the STL-10 dataset. Experiments with these dataset can be found in the following four papers (among others):

Dai and Van Gool (2013) "Ensemble Projection for Semi-supervised Image Classiļ¬cation"
Goodfellow et al. (2012) "Large-Scale Feature Learning With Spike-and-Slab Sparse Coding"
Coates et al. (2011) "An Analysis of Single-Layer Networks in Unsupervised Feature Learning"
Coates and Ng (2011) "Selecting receptive fields in deep networks"
Summary: This is an ok paper about an important topic. The approach the authors take is interesting and I think this work could have an impact in the field. Some relevant previous work has been neglected both from the literature review and from the empirical comparisons.

Submitted by Assigned_Reviewer_13

This paper explores semi-supervised learning with deep generative models. It concentrates on recent developments by Kingma and Welling and Rezende et al. which formulate an explicit model of the data density and inference models based on deep neural networks. A form of scalable variational inference is applied as well as a stochastic variant of backpropagation. The paper presents two alternative approaches: the first, learning a latent variable generative model, and then using samples from the posterior to train a classifier; the second trains a class-conditional generative model and applies a classification-as-inference methodology.

Quality

I thought this was a high-quality paper; it leverages recent results in variational inference and generative models and applies them to an important domain. The paper is readable and the results are convincing. One weak point is that it seeems that the paper was rushed to submission; for example why are there no M2 results for the SVHN and NORB datasets? Also, if I understood correctly, all results except the proposed method in Table 1 are taken from Rifai et al. So there aren't a lot of experimental results reported that were conducted by the authors.

Clarity

The variational inference method presented is highly technical and therefore 3.1-3.3 are involved; however, I thought that the material was presented in a very clear manner. Algorithms (1 and 2) supplement the math. The authors are careful about which details to present and which to hold back.

Originality

The paper leverages much from Kingma and Welling and Rezende et al. However, the development of the methods for semi-supervised learning (in particular, Method 2) is novel work and considerably different than other semi-supervised learning frameworks that I know of. As pointed out in the conclusions, the formulation of the augmented loss (8) is clunky -- though this approach has been used a lot in training deep architectures with multiple objectives.

Significance

I think this paper is significant because the particular generative framework employed is interesting. It can leverage future developments in deep learning. The experiments demonstrate that it is competitive against other semi-supervised learning techniques.

Specific comments

p2, first paragraph of section 2 "to improve upon the classification performance that would be obtained using the unlabelled data alone" -> you mean labelled, right?

p5, "We also we the same" ???

p7, The caption for Figure 2 should explain what's being presented in the figure (even if there is a description in the main body text); it's not obvious

Summary: This paper extends recent work on variational inference on deep generative models to the semi-supervised setting. It is well motivated and well executed though it could be more polished.

Submitted by Assigned_Reviewer_41

The paper applies a set of recent state-of-the-art techniques for variational Bayes, inference networks, and stochastic backpropagation to the problem of semi-supervised learning. The combination of these techniques provides an efficient and scalable Bayesian framework for solving the problem.

The authors propose two variants of their models called M1 and M2. The first one builds an unsupervised model first, and then, trains a classifier on the learned latent representation. The second one learns a joint generative model where the labels are part of it.

The proposed model is evaluated empirically on several well-known datasets (MNIST, NORB, SVHN) show to produce more accurate classifier than previous methods and other baselines. I am not exactly sure why only results for M1 are shown and not those for M2 on the last two datasets. Additional visualizations are provided to illustrate the disentanglement between the class and the "style" of the sample as modelled by the generative model.

Minor comments:

- It seems that there is an inconsistency between the use of p(x|f(z)) used in Equation 1 and p(x|z) in Equation 3. Same for f(y,z) in Equation 2.

- A related paper is: "Larochelle and Bengio, 2008. Classification using Discriminative Restricted Boltzmann Machines". They use of similar trade-offs between generative and discriminative objective as the one defined in Section 3.1, Equation 8.

- l.200: where R(x) -> where R(x)R(x)^T ?

- l.261: variable g is not explicitly defined
Summary: While the paper does not bring a substantially new machine learning insight on the semi-supervised problem, the paper is well written and provides a sound and clearly designed framework to apply Bayesian classifiers to semi-supervised tasks.
Author Feedback
Author rebuttal: Reviewers, thank you for you time, appreciations and helpful comments.
Your feedback will, naturally, be incorporated in the final version of the paper.
Here is our response to the main points of your critique.

=== REVIEWER 13:
“One weak point is that it seems that the paper was rushed to submission; for example why are there no M2 results for the SVHN and NORB datasets? Also, if I understood correctly, all results except the proposed method in Table 1 are taken from Rifai et al. So there aren't a lot of experimental results reported that were conducted by the authors.“

While we believe that the current results already sufficiently demonstrates the advantages of our model over earlier approaches, the final paper will be made even stronger with regards to experimental section.

- “p2, first paragraph of section 2 "to improve upon the classification performance that would be obtained using the unlabelled data alone" -> you mean labelled, right?

Yes, indeed.

=== REVIEWER 2:
- “On the negative side, the specific two proposed approaches seem fairly straightforward. This isn't a strong criticism but it does impact the novelty of the paper. It would have been better if the experiments really focused on the differences between the two, in order to give the reader some idea of their differential abilities, but that wasn't really the case as they were only really compared directly in one set of experiments limited to the MNIST hand-written digits database.”

Both M1 and M2 constitute generative models (VAEs) parameterized with multilayered neural networks. As explained in the paper, the difference is that M1 assumes a single top-level latent vector (z) while in M2 the class label is also considered a top-level variable (which is latent for a subset of the datapoints) from which x is generated. In M1, the classifier takes as input the learned ‘z’ representation. We will furtherly clarify the distinction between the two approaches in the final version.

- “More seriously, there are a large number of similar methods that
are not discussed and not compared empirically. In particular there are many papers not cited discussing learning in the self-taught learning framework (including the original self-taught learning paper of Raina et al. (2007)) which is essentially identical to the proposed M1 approach. Other missing citations and comparisons are discussed below in more specific comments on the experiments.”

We thank the reviewer for these additional references. The problem setup in the self-taught framework generally assumes an unlabeled set that is drawn from a different distribution than the labeled set, which differs a bit from our the semi-supervised learning setup where the underlying distributions are assumed equal. But self-taught learning methods can certainly be applied to semi-supervised learning problems, and interestingly enough our M1 approach could be applied to self-taught learning problems. We will discuss the relationship and include an experimental comparison in the final version.

- “2 - Conditional Generation: [...] These results are mildly compelling in that they capture -- for SVHN -- the color information of both the background and foreground but do not seem to capture the font style. [...]"

The MNIST font style is certainly captured, so this critique is probably due to an mis-interpretation by the reviewer of figure 2. The figure is admittedly too small and will be enlarged. The left-most column consists of a random sample of hand-written digits from the test set. The other columns are analogies generated from the learned model p(x|z,y). The writing style (‘z’) of each generated row is inferred from the test-set image in the left-most column, while the class label (‘y’) iterates from [0,...,9] from left to right. We think this success is quite exciting, and that it accentuates a big advantage of the fully probabilistic approach: the ability to perform meaningful inferences on which the model was not explicitly trained.

- “1 - Benchmark Classification. [...] The first is a basic comparison of the proposed approach in a semi-supervised version of the MNIST hand-written digits database. [....] I think it's important to see how the proposed methods (M1 and M2) work in the larger labeled set regime. [...]
-3 - Image Classification: The third set of experiments explore semi-supervised classification with SVHN and NORB image datasets. The authors state that there are no published results on these datasets in the semi-supervised setting so they compare to their own results using nearest neighbors and Transductive SVM. While I know of no results on these two datasets in the semi-supervised regime, there are plenty of results on other image classification datasets with published results in the semi-supervised regime.[...]”

We would like to thank the reviewer for these suggestions. We believe that the current results already provide sufficient evidence for the effectiveness of our approach. However, to make the final paper even stronger we will incorporate experimental results on larger sets of labeled data (for MNIST), and also include comparisons with other datasets mentioned by reviewer 2.